# Alternative Splicing Regulation of Low-Frequency Genetic Variants in Exon 2 of *TREM2* in Alzheimer’s Disease by Splicing-Based Aggregation

**DOI:** 10.3390/ijms22189865

**Published:** 2021-09-13

**Authors:** Seonggyun Han, Yirang Na, Insong Koh, Kwangsik Nho, Younghee Lee

**Affiliations:** 1Department of Biomedical Informatics, University of Utah School of Medicine, Salt Lake City, UT 84108, USA; seonggyun.han@utah.edu; 2Transdisciplinary Department of Medicine and Advanced Technology, Seoul National University Hospital, Seoul 03080, Korea; yirangna@snu.ac.kr; 3Department of Physiology, Hanyang University, Seoul 04763, Korea; 4Department of Radiology and Imaging Sciences, Indiana Alzheimer’s Disease Research Center, Indiana University School of Medicine, Indianapolis, IN 46202, USA; 5Center for Computational Biology and Bioinformatics, Indiana University School of Medicine, Indianapolis, IN 46202, USA

**Keywords:** *TREM2*, Alzheimer’s disease, alternative splicing, low-frequency variant, aggregation of low-frequency variants

## Abstract

*TREM2* is among the most well-known Alzheimer’s disease (AD) risk genes; however, the functional roles of its AD-associated variants remain to be elucidated, and most known risk alleles are low-frequency variants whose investigation is challenging. Here, we utilized a splicing-guided aggregation method in which multiple low-frequency *TREM2* variants were bundled together to investigate the functional impact of those variants on alternative splicing in AD. We analyzed whole genome sequencing (WGS) and RNA-seq data generated from cognitively normal elderly controls (CN) and AD patients in two independent cohorts, representing three regions in the frontal lobe of the human brain: the dorsolateral prefrontal cortex (CN = 213 and AD = 376), frontal pole (CN = 72 and AD = 175), and inferior frontal (CN = 63 and AD = 157). We observed an exon skipping event in the second exon of *TREM2*, with that exon tending to be more frequently skipped (*p* = 0.0012) in individuals having at least one low-frequency variant that caused loss-of-function for a splicing regulatory element. In addition, genes differentially expressed between AD patients with high vs. low skipping of the second exon (i.e., loss of a *TREM2* functional domain) were significantly enriched in immune-related pathways. Our splicing-guided aggregation method thus provides new insight into the regulation of alternative splicing of the second exon of *TREM2* by low-frequency variants and could be a useful tool for further exploring the potential molecular mechanisms of multiple, disease-associated, low-frequency variants.

## 1. Introduction

Alzheimer’s disease (AD) is one of the most common neurodegenerative disorders leading to cognitive impairment in the elderly. AD is characterized by accumulation of toxic amyloid-β (Aβ) plaques and neurofibrillary tau tangles along with synaptic deficits, ultimately resulting in memory loss and cognitive decline. Importantly, twin and family studies have shown that the inheritance of genetic factors plays a significant role in the development of AD [1]. In addition, recent large-scale, genome-wide association studies have identified genetic factors associated with increased risk of developing AD [2,3,4]. The triggering receptor expressed in myeloid cells 2 (*TREM2*) has become one of the most well-known genes implicated in increased risk of late-onset AD [5]. *TREM2* encodes a membrane protein that features an immunoglobulin extracellular domain expressed in myeloid cells, including microglia in the brain [6]. It mediates the phagocytosis of apoptotic neurons, inflammatory responses, and microglia proliferation [5,7,8,9,10], and so strongly affects risk of AD. In fact, *TREM2* contributes to the building-up of Aβ plaque and accumulation of tau tangles in the AD brain [10,11,12,13].

Recent studies have identified low-frequency variants within *TREM2* as being significantly associated with AD risk [14,15,16,17,18,19,20], including rs142232675, rs143332484, rs2234256, rs2234253, and rs2234255. In particular, a neuropathological study of rs75932628 validated that it increases the risk of developing late-onset AD by almost three times, which is similar to having one copy of the *APOE* ε4 allele [19,21]. Another variant, *TREM2* R47H (i.e., rs75932628), has been shown to result in a loss-of-function phenotype [22,23] and decreased *Trem2* mRNA and protein levels in mice [24]. It specifically impairs the binding functions of *TREM2* in vivo through inducing structural changes [23], and this loss-of-function could reduce microglial activation, thereby affecting Aβ clearance in the brain, promoting AD pathogenesis [25]. However, functional roles—the molecular mechanisms by which a variant is implicated in producing the end-point phenotype—still remain to be elucidated for most disease-associated *TREM2* variants [26]. In terms of the statistical aspect of annotating functionally unknown genetic variants, it remains difficult to investigate the potential molecular functions of low-frequency variants (MAF < 5%) because such exploration requires very large sample sizes [27]. Studies have been undertaken to investigate low-frequency variants (e.g., somatic or rare variants) by means of aggregation at gene or pathway levels with phenotypes at the association level [28,29]; however, the intervening functional mechanisms of those variants still remain unexplored.

Increasing evidence suggests that splicing patterns are a key factor for *TREM2* function, and its aberrant splicing imparts AD risk [24,26]. A recent study showed that a soluble form of TREM2 (sTREM2) with skipping of the fourth exon (i.e., protein lacking the transmembrane domain) has increased expression in AD patients and could be used as a biomarker for AD [26]. In addition, an AD risk variant was observed to promote mis-splicing of the second exon of *Trem2* in mice [24]. Notably, most AD risk variants in *TREM2* are located in its second exon, which encodes the Ig-like V type domain—including sites of interactions between TREM2 and its ligands, such as the important AD-associated factors of ApoE, HDL, LDL, Aβ, and clusterin [5,26,30,31]. Although the exact ligands activating TREM2 are yet unclear, the second exon is essential for TREM2 function, especially ligand binding, and these reports suggest that variants in the second exon can lead to loss-of-function by affecting interactions between TREM2 and its ligands [5].

We previously proposed a splicing decision model for identifying SNPs that impact splicing regulatory elements (SREs), which were predicted on the basis of hexameric sequence motifs [32]. SREs are *cis*-acting splicing elements that recruit RNA-binding proteins (RBPs) that allow the spliceosome to effectively and correctly recognize the exon-intron boundary, thereby producing accurate splicing [33,34]. Our splicing decision model can predict loss-of-function of SREs that is caused by genetic variants [32].

Working from our assumption that low-frequency variants implicated in SRE sites may affect *TREM2* function by influencing splicing of the second exon, our study sought to aggregate variants on the basis of their impact on splicing mechanisms, and then to apply this method to elucidate the functional roles of low-frequency variants in *TREM2* exon two. We analyzed RNA-seq and whole genome sequencing (WGS) data generated from two independent cohort studies in the Accelerating Medicines Partnership Alzheimer’s Disease (AMP AD) project, namely the ROS and MAP (ROSMAP) [35] and Mount Sinai Brain Bank (MSBB) [36] projects. Three frontal brain regions were represented in this data: the dorsolateral prefrontal cortex (DLPFC), frontal pole (FP), and inferior frontal (IF).

## 2. Results

### 2.1. The Second Exon of TREM2 Is Skipped in All Three Brain Regions

Within the splicing unit, which constitutes those genome regions locally affecting splicing of the second exon, we found 64 genetic variants, of which 63 were low-frequency (MAF < 5%) (Appendix A). As described in Figure 1, the TREM2 gene is recorded in the Ensemble GRCh37.75 (hg19) annotation reference dataset as having three transcript isoforms, ENST00000373122, ENST00000338469, and ENST00000373113, among which exon two is a constitutive exon (i.e., no AS event). We first estimated the percent spliced in (PSI) of the second exon for each individual on the basis of RNA-seq data (Figure 1). PSI refers to the fraction of exon inclusion, ranging from zero to one, with zero indicating skipping in all transcripts (all skipping junction reads) and one indicating inclusion in all transcripts (all inclusion junction reads).

We observed a skipping event of the second exon of TREM2 in all three brain regions of both the independent cohorts (Figure 2); the mean PSI levels were 0.898, 0.904, and 0.913 for the DLPFC, FP, and IF, respectively.

### 2.2. Aggregation of Low-Frequency Variants Potentially Affecting Inclusion of the 2nd Exon 

As described in Figure 1b, we developed a splicing-guided aggregation method in order to collect multiple low-frequency loci potentially affecting splicing regulation (i.e., leading to loss-of-function of an SRE). In the splicing unit for the second exon, we identified 63 low-frequency genetic variants (the red and blue dots at the top of Figure 3a). Among those, ten variants (the red dots in Figure 3a) were associated with predicted loss of SRE function (i.e., changed the hexameric sequence of an SRE), and so could potentially impact exon two splicing level. The genomic position information of ten variants within SREs is described in Appendix A. Two of those ten variants (rs143332484 and rs201258314) are located in the second exon, and eight in the flanking introns; the genomic positions and frequencies of the variants are listed in Figure 3b. We aggregated individuals with at least one of ten loss-of-function variants as the functional variant group (described in Appendix A), while individuals without any of the ten variants were grouped into a non-functional variant group.

With those defined groups, we then evaluated statistically whether the aggregated low-frequency variants influence splicing (Figure 4). We found that for the functional variant group, PSI levels of the second exon tend to decrease in the frontal brain region, *p* = 0.0012, relative to the non-functional variant group; thus, the second exon is more frequently skipped in the functional variant group (the right boxplot in Figure 4b). However, there was no significant difference in PSI between the combined low-frequency variant groups (i.e., all low-frequency variants regardless of functional impact on SREs) and a non-variant group (i.e., samples without any low-frequency variants). That is, these findings suggest that our splicing-guided aggregation method that aggregates only functional variants could be a crucial tool for elucidating how at least some low-frequency variants potentially exert their function. We validated our results with a random permutation test, which randomly selected a subset (*n* = 52) from the variant group (*n* = 224) and treated them as the functional variant group in the linear regression, carried out 1000 times (Figure 4c).

### 2.3. Dependence of Second Exon Skipping on TREM2 Variants

Since it has been reported that cognitively normal older adults (CN) and AD patients differ significantly in TREM2 protein structure [26], we evaluated whether skipping of TREM2 exon two is associated with AD. We did not find any significant difference in exon skipping rates between the CN and AD groups (*p* = 0.7184). Our next step was to determine the association of exon skipping with functional and non-functional variants in the context of CN and AD. As mentioned above, we observed that when considering all individuals (i.e., CN plus AD), skipping of the second exon tends to be more frequent in the functional variant group compared to the non-functional variant group. We also found that the splicing rate of the functional variant group is likely to increase for AD patients but not CN individuals (Table 1).

### 2.4. Functional Roles of Genes Differentially Expressed between Those Having High and Low Rates of Second Exon Skipping 

Since the rate of second exon skipping tends to increase (i.e., PSI levels decrease) in the functional variant group relative to the non-functional variant group, we further performed differential gene expression analysis comparing individuals with high skipping rates and those with low skipping in order to identify downstream pathways impacted by the exon skipping event (i.e., loss of the functional domain of TREM2). We identified 65 differentially expressed genes (Appendix A) having FDR < 0.05 and fold change > 2. Gene set enrichment analysis on the identified genes revealed enrichment of immune-related functional pathways, including “Immune system”, “Interluekin-1 signaling”, and “leukocyte activation” (Appendix A).

## 3. Discussion

Here, we proposed a splicing-guided aggregation approach and applied it as a case study to functionally annotate ten low-frequency variants located in and closely around the second exon of *TREM2* in AD patients. *TREM2* and its low-frequency variants are associated with the pathogenesis of neurodegenerative diseases such as AD and Parkinson’s disease [7]. The *TREM2* gene encodes an immunoglobulin receptor that is implicated in immune and inflammatory pathways related to AD progression [5,7]. Moreover, TREM2 is connected to Aβ plaque structure and the aggregation of hyper-phosphorylated tau, hallmarks of AD progression [8,10,13]; indeed, *TREM2* has been highlighted as the target of a novel therapeutic strategy for AD progression [9]. Low-frequency variants in *TREM2* have been reported to increase AD risk by almost 3–4 times, and, therefore, investigating the functional roles of such variants may be crucial to understanding the involvement of TREM2 in various neurodegenerative diseases [26].

The nature of low-frequency variants limits our ability to statistically detect their mechanistic effects (e.g., gene expression and AS changes). We hypothesized that our splicing-guided aggregation approach may offer an opportunity to detect the collective effect of multiple low-frequency variants. We demonstrated that patients having at least one of ten low-frequency variants within SRE sites (rs182653531, rs115121185, rs184276085, rs936961326, rs143332484, rs201258314, rs565502230, rs59377666, rs74390253, and rs369487317) tend to have increased skipping of the second exon of *TREM2* (Figure 4). Nine of these variants had minor allele frequencies of less than 0.01 in the datasets we analyzed (the exception being rs143332484). Thus, our splicing-guided aggregation approach would provide a useful tool for investigating the collective functional mechanism of low-frequency variants, and so could provide new insights into understanding their role in disease.

TREM2 is a transmembrane protein that interacts with ligands such as ApoE, HDL, LDL, Aβ, and clusterin (CLU) [5]. Our present focus, the second exon, encodes the binding site of the Ig-like V type domain needed for such interactions, and, thus, skipping of that exon impacts TREM2 protein function through loss of a functional domain. Interestingly, the second exon is enriched in AD-associated variants [26]. In vivo studies using a mouse model of AD have shown one second-exon SNP, rs75932628, to lead to the loss of TREM2 function with consequent increased neuritic dystrophy and plaques [22]. Furthermore, aberrant splicing patterns of *TREM2* have been shown to result in abnormal protein function that supports AD development [24,26]. Therefore, our observation of AS of the second exon may enable us to elucidate the biological and mechanistic relationship between *TREM2* and its variants in AD progression.

While we did not confirm the alternative splicing event of the second exon in our dataset through experimental validation, we did observe skipping in two independent cohort sets representing three brain regions, and, interestingly, a recent study experimentally validated that the second exon is also skipped in the human brain using qRT-PCR and Western blot [37]. In addition, rs75932628 has been reported to lead to mis-splicing of the second exon in mice [24], which could be evidence supporting the research showing that the exon can be skipped in the human brain as well. However, alternative splicing of the second exon was not observed with rs75932628 in humanized *TREM2* R47H knock-in mice [24]. Therefore, a further experimental study may be required to validate the second exon skipping event in the brains of human AD patients.

Our approach concerning the annotation of a variant as functional or non-functional with respect to splicing stems from the established fact that genetic variants present in splicing regulatory elements (SREs) can affect splicing efficiency by altering the binding affinity of a splicing factor, resulting in alternative splicing events. Thus, SRE locations serve as a potential criterion for defining the functionality of a given SNP. We utilized a genome-wide scan of SRE locations and mapped the overlap of variants with SRE regions to classify a variant as functional. As expected, we observed that patients having at least one functional variant predicted to lead to loss of SRE function showed increased skipping of the second exon, but the group of patients having any rare variant did not exhibit increased exon skipping (Figure 4b).

A previous study reported that variants of *TREM2*, especially within the second exon, do not affect its transcript expression, suggesting they might instead influence a post-transcriptional process [26]. Of our ten aggregated low-frequency variants, rs143332484 is a well-known variant for increasing AD development and tends to decrease TREM2 levels in the cerebrospinal fluid (CSF) [5,26,38,39,40,41]. In this context, our results suggest that rs143332484 may decrease CSF TREM2 by increasing skipping of the second exon but not transcript expression. However, no significant difference in PSI of the second exon was observed between CN and AD patients. Another prior study also reported a similar result of no evident difference between CN and AD in terms of overall *TREM2* expression [26]. These findings may suggest the presence of another hidden factor beyond splicing that is involved in the impact of multiple low-frequency variants on AD pathogenesis.

We further investigated the prospect of downstream functional impact of second exon skipping in *TREM2*. As the second exon encodes the Ig-like V type domain, which is important for interaction between TREM2 and its ligands, skipping of that exon could affect downstream signaling [5]. Gene set enrichment analysis of the genes differentially expressed between individuals having low and high skipping of the second exon revealed significant enrichment of immune-related pathways and GO terms. This result is consistent with prior knowledge of TREM2, which is well-documented, to be important for immune function in AD [5,7,8,42]. In fact, a recent experimental study demonstrated that the second exon skipping event could change immune function, especially the IFN-I response, in the human brain [37]. Thus, while we observed no significant difference in the rate of second exon skipping in CN and AD patients having ten aggregated low-frequency variants, immune-related functions may be particularly affected in AD patients with the low-frequency variants. As such, further analysis is merited to identify associations of this skipping event with clinical features of AD.

As we observed a differential skipping rate between functional and non-functional variant groups in AD but not in CN, we investigated if AD individuals carry more loss-of function variants than CN. As shown in Appendix A, seven AD individuals (19.4% of AD) carried more than one loss-of-function variant in AD, while only one CN individual (6.3% of CN) carried more than one loss-of-function variant.

However, this study possesses several limitations, and these should be raised in order to avoid over-interpretation. For example, we did not validate the novel second exon skipping event by performing experiments or analyzing proteomics data. While we did look into samples having matched proteomics data, unfortunately, there is little information on TREM2 in the data. Another limitation is that we applied our splicing-guided approach to *TREM2* alone; with AD being a complex disease, further analysis of AD risk genes or the whole genome using our method needs to be performed. Finally, we considered only SRE information as a potential factor impacting splicing, neglecting several other known factors such as the distance of variants from a splicing site, RNA secondary structure, and spliceosome expression. A splicing-guided aggregation method that more comprehensively incorporates such factors can help us more effectively annotate the functional roles of low-frequency variants.

## 4. Materials and Methods

### 4.1. Analysis of RNA-Seq and WGS Data

Demographic information of analyzed samples and brain regions is described in Appendix A. RNA-seq (BAM files) and WGS data (VCF files) were downloaded from the Synapse database (www.synapse.org (accessed on 1 July 2019)). Data was obtained from the Accelerating Medicines Partnership® Alzheimer’s Disease (AMP AD) project, which includes two independent cohorts: the ROS and MAP (ROSMAP) project [35] and the Mount Sinai Brain Bank (MSBB) [36]. RNA-seq was performed in three frontal brain regions of cognitively normal older adults (CN) and AD patients: the dorsolateral prefrontal cortex (DLPFC), frontal pole (FP), and inferior frontal (IF). In order to ensure matching reference versions and consistent mapping of reads, we converted the BAM files to FASTQ files using SAMtools [43] and aligned the RNA-seq reads to the human reference genome (GRCh37.75 based on the hg19 reference) using STAR [44] for each participant. Then, we employed rMATs to identify alternative splicing events and quantified their splicing rate as percent spliced in (PSI) [45], which means the fraction of transcripts including an alternative splicing exon. The WGS data was previously preprocessed and analyzed in a joint study of ROSMAP and MSBB and provided as VCF files. We extracted the genotypes of each variant in TREM2 for each sample from those files using the readVcf function of the VariantAnnotation R/Bioconductor package [46].

### 4.2. Definition of Splicing-Guided Aggregation and Functional Variants

We implemented our previously developed computational splicing decision model to define “functional variants” potentially affecting the pattern of 2nd exon skipping [32]. In this decision model, low-frequency variants are collected according to the splicing unit, here a genomic region consisting of the 2nd exon, flanking introns, and neighboring exons (red box in Appendix A); these are regions in which genetic variants can locally affect splicing of the 2nd exon. Then, we define “functional variants” according to their overlap with a cis-regulatory element of the splicing machinery, the hexameric splicing regulatory element [47] (SRE, i.e., exonic splicing enhancer [ESE], exonic splicing silencer [ESS], and intronic splicing enhancer [ISE]); these sequence motifs facilitate proper splicing as described in detail in our previous study [34]. We scanned the splicing unit for variants and checked if the hexameric sequences surrounding those variants perfectly matched SRE sequences according to our previously published method [32,48]. We then defined a given variant as “functional” if the variant lost the SRE sequence in its alternative allele, implying the loss of SRE function and, therefore, a potential effect on splicing (Appendix A). We defined samples having low-frequency variants as the “variant group”, and those without any variants as the “non-variant group”. We additionally defined samples having at least one of functional low-frequency variants as the “functional variant group”, and those not having any functional low-frequency variants, including those with no variants, as the “non-functional variant group”.

### 4.3. Statistical Evaluation of Aggregated Low-Frequency Variants in TREM2 Exon 2

Given the goal of evaluating the potential and applicable ability of our aggregation methods in terms of the functional annotation of low-frequency variants in a specific context, namely exon 2 of TREM2, we used linear regression to compare the functional and non-functional variant groups in terms of exon 2 AS level (PSI). We specifically carried out mega-analysis, which combines patient data for all three brain regions in order to increase the sample size and statistical power. The resulting *p*-values were validated through the determination of empirical *p*-values using a permutation test in which “functional variant” status was randomly assigned to members of the variant group 1000 times, and then the linear regression was repeated and *p*-values compared.

### 4.4. Functional Implication of TREM2 Exon 2

We defined samples with PSI levels below the first quantile (PSI = 0.85) and greater than the third quintile (PSI = 1) as low-skipping and high-skipping groups, respectively. We then conducted a non-parametric test, the Wilcoxon rank sum test, on genes to identify differentially expressed genes (DEGs) between the two groups. The statistical cutoff for identifying DEGs consisted of a false discovery rate (FDR)-corrected *p*-value < 0.05 and fold change > 2. We further performed pathway over-representation analysis using ConsensusPathDB (CPDB) [49] including Reactome database [50,51] and GO terms to identify functional pathways enriched among the identified DEGs. We defined significant enrichment terms as those having FDR q < 0.05. Significantly enriched immune-related functional pathways and GO terms manually were selected to visualize.

## 5. Conclusions

In conclusion, the splicing-guided aggregation method could be a useful tool for investigating the potential functional roles of low-frequency variants. In addition, our results provide new insight into the molecular mechanisms by which low-frequency variants impact the second exon of *TREM2*.

## Figures and Tables

**Figure 1 ijms-22-09865-f001:**
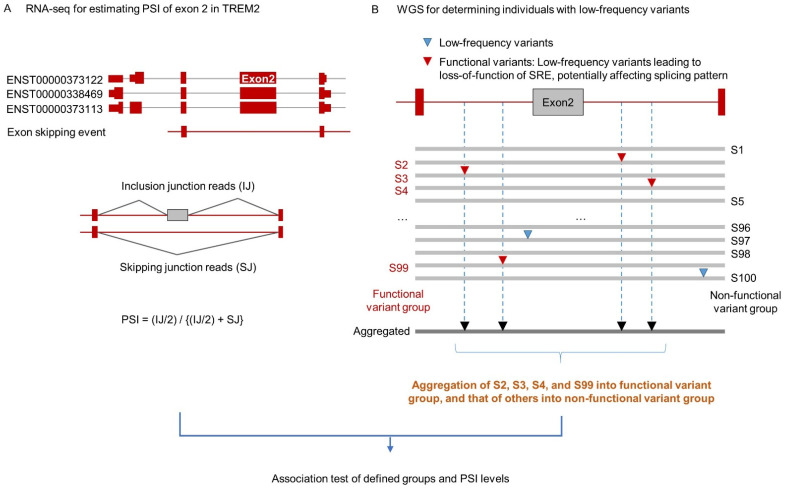
(**A**) The three transcript isoforms of *TREM2*, according to annotation of the hg19 reference genome. The rate of exon 2 skipping could be estimated from inclusion and skipping junction reads as PSI. (**B**) Splicing-guided aggregation of rare variants within the *TREM2* 2nd exon, flanking introns, and neighboring exons. Blue reverse triangles indicate low-frequency variants not leading to SRE loss-of-function; red triangles denote low-frequency functional variants predicted to lead to SRE loss-of-function (see Materials and Methods). Samples with these functional low-frequency variants are aggregated into a functional variant group, while those lacking such variants comprise a non-functional variant group (see Materials and Methods).

**Figure 2 ijms-22-09865-f002:**
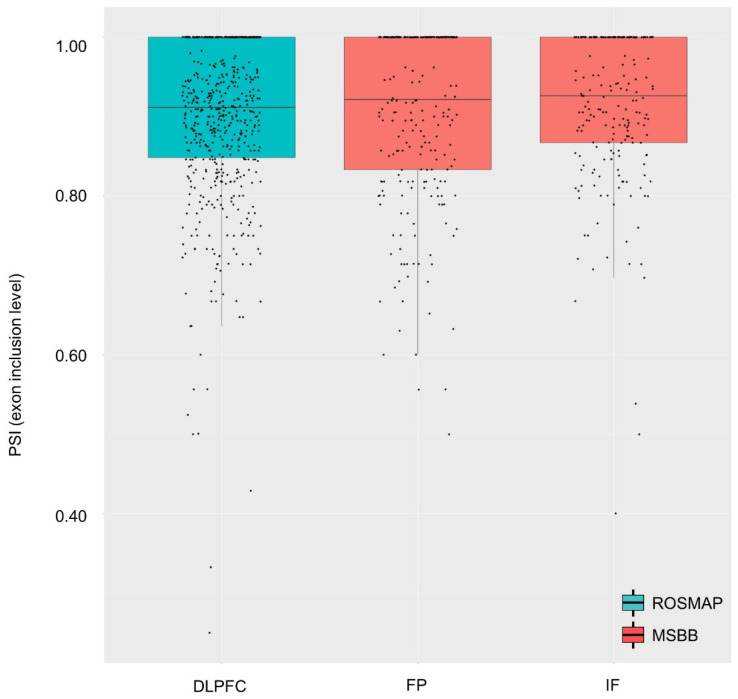
Distribution of *TREM2* 2nd exon PSI values for each investigated brain region. *X*-axis and *Y*-axis indicate PSI and brain regions, respectively. Cyan and red boxplots were respectively generated from ROSMAP and MSBB cohort datasets.

**Figure 3 ijms-22-09865-f003:**
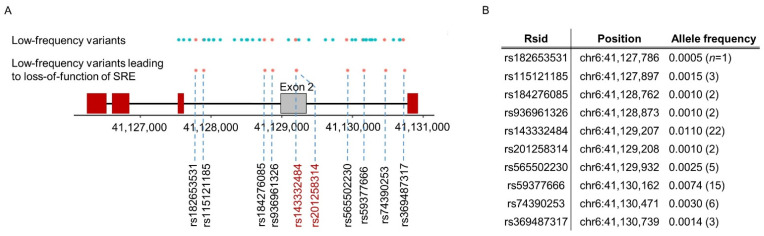
(**A**) Schematic depicting *TREM2* exons and low-frequency variants within the exon 2 splicing unit. Cyan dots are low-frequency variants without predicted splicing impact, and red dots are low-frequency variants leading to loss of SRE function (functional variants). (**B**) Details of low-frequency variants potentially affecting exon 2 splicing.

**Figure 4 ijms-22-09865-f004:**
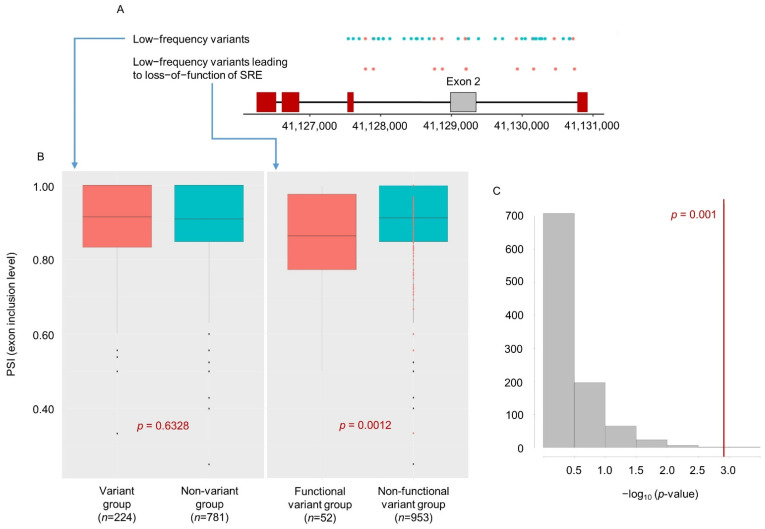
(**A**) Schematic of *TREM2* exon positions and low-frequency variants found within exon 2, flanking introns, and neighboring exons. (**B**) Distribution of PSI values among groups defined according to the presence and predicted splicing functionality of low-frequency variants. The left boxplot compares the variant and non-variant groups with the variant group comprising all samples with low-frequency variants (the blue and red dots in **A**); the right boxplot compares the functional variant group, samples with low-frequency variants leading to loss of SRE function, and the non-functional variant group, samples with either no variants or low-frequency variants not leading to loss of SRE function. There was no significant difference in PSI between the variant and non-variant groups (left boxplot), but a difference was found between the functional and non-functional variant groups (right boxplot). Red points in the right boxplot indicate the samples having low-frequency variants not leading to loss of SRE function (i.e., samples in the variant group but not the functional variant group). (**C**) Histogram of *p*-values generated from 1000 random permutations selecting samples (*n* = 52) to constitute the functional variant group. Only five instances obtained with random selection were the *p*-values less than our original result (i.e., 0.0012).

**Table 1 ijms-22-09865-t001:** Statistical difference between groups with and without SRE-impacting low-frequency variants.

	Frontal Region (DLPFC, FP, IF)
Functional variant group vs. Non-functional variant group	0.0012 (52 vs. 953)
Functional variant vs. Non-functional variant in AD	0.0007 (36 vs. 644)
Functional variant vs. Non-functional variant in CN	0.3592 (16 vs. 309)

## Data Availability

WGS data for ROSMAP and MSBB: https://www.synapse.org/#!Synapse:syn11707419 (accessed on 1 September 2021). RNA-seq data for ROSMAP: https://www.synapse.org/#!Synapse:syn4164376 (accessed on 1 September 2021). RNA-seq data for MSBB: https://www.synapse.org/#!Synapse:syn7416949 (accessed on 1 September 2021).

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
