# Peer review of "Alternative Splicing Regulation of Low-Frequency Genetic Variants in Exon 2 of TREM2 in Alzheimer’s Disease by Splicing-Based Aggregation"

_ijms, 2021, doi:10.3390/ijms22189865_

Round 1

Reviewer 1 Report

In their manuscript entitled “Alternative splicing regulation of low frequency genetic variants in exon 2 of TREM2 in Alzheimer’s disease by splicing-based aggregation” the authors use bioinformatics tools to identify loss-of-function variants in splicing regulatory elements of TREM2 exon 2. This targeted interrogation of already existing data from GWAS and RNAseq studies is an excellent example of the need and usefulness of this kind of studies.

However, I have some concerns, especially in the RESULTS section:

  1. A subsection within the Results-Section showing the number, sequence and position of the identified SREs within the analysed splicing unit should be included. The exact position of the 10 SNPs located within these structures should be also indicated.
  2. The sentence in lines 131-133 should be rewritten since it is not clear if individuals with the 10 loss-of-function variants contain all of these or at least one. It has been explained in Materials and Methods, but should be also clear in this sentence.

I would also suggest to add a Table (might be supplemental) that includes all 48 subjects carrying loss-of-function variants identifying who are CN and who are AD. All SNPs should be also shown, so that individuals carrying more than one variant can be easily identified.

  1. Lines 138-142: Although the results are shown in Figure 4B, the PSI values should be also given in the text.
  2. Line164-165: please, add a reference to the statement that TREM2 structure differs between AD patients and controls.
  3. The authors report, that the splicing rate of the functional variant group is increased in AD compared to CN individuals. Is there a difference in the location of the variants detected? Do AD individuals carry more than one loss-of-function variant?
  4. Sentence between lines 182 and 183: Table S2 is missing, but should be added. Did the authors analyse the interaction of the 65 proteins by STRING?

Minor concerns:

  1. Line 46: “It” should be deleted.
  2. Figure 3B: “Allele frequency” should be substituted by “MAF”
  3. “AS” in lines 223, 288, 289 should be spelled out.
  4. Lines 278-279: There is no Table 1 in the manuscript, but TableS1 in the Supplemental Material, please rectify.
  5. Genotype frequencies of APOE genotypes should be shown (in brackets after the number) in Table S1.
  6. Figures S2 and S3: The meaning of the red vertical line should be explained in the Figure captions.

Reviewer 2 Report

Han et al. evaluate if a splicing-guided aggregation method could be a useful tool for investigating the potential functional roles of low-frequency variants in exon 2 of TREM2 in Alzheimer’s disease.

The manuscript is well done written and presented. The aim of the manuscript is very important, as the authors highlighted. The Introduction and the Materials and Methods chapters are clear and comprehensible. The Results and Discussion chapters are well presented and evaluated.

Round 2

Reviewer 1 Report

The authors have addressed my concerns adequately, and the overall quality has been improved. For my understanding, the manuscript can be published in its current form.